# Skp2 and Slug Are Coexpressed in Aggressive Prostate Cancer and Inhibited by Neddylation Blockade

**DOI:** 10.3390/ijms22062844

**Published:** 2021-03-11

**Authors:** Alena Mickova, Gvantsa Kharaishvili, Daniela Kurfurstova, Mariam Gachechiladze, Milan Kral, Ondrej Vacek, Barbora Pokryvkova, Martin Mistrik, Karel Soucek, Jan Bouchal

**Affiliations:** 1Department of Clinical and Molecular Pathology, Faculty of Medicine and Dentistry, Palacky University and University Hospital, 779 00 Olomouc, Czech Republic; alena.mickova@upol.cz (A.M.); daniela.kurfurstova@seznam.cz (D.K.); marjammg@gmail.com (M.G.); 2Institute of Molecular and Translational Medicine, Faculty of Medicine and Dentistry, Palacky University, 779 00 Olomouc, Czech Republic; martin.mistrik@upol.cz; 3Department of Urology, University Hospital, 779 00 Olomouc, Czech Republic; kral.milan@seznam.cz; 4Department of Cytokinetics, Institute of Biophysics of the Czech Academy of Sciences, 612 65 Brno, Czech Republic; vacek@ibp.cz (O.V.); ksoucek@ibp.cz (K.S.); 5International Clinical Research Center, Center for Biomolecular and Cellular Engineering, St. Anne’s University Hospital in Brno, 602 00 Brno, Czech Republic; 6Department of Experimental Biology, Faculty of Science, Masaryk University, 625 00 Brno, Czech Republic; 7Department of Genetics and Microbiology, Faculty of Science, Charles University, BIOCEV, 252 50 Vestec, Czech Republic; barbora.pokryvkova@natur.cuni.cz

**Keywords:** prostate cancer, Skp2 (S-phase kinase-associated protein 2), Slug, immunohistochemistry, multiplex, neddylation

## Abstract

Prostate cancer (PCa) is the second leading cause of cancer-related deaths in men in Western countries, and there is still an urgent need for a better understanding of PCa progression to inspire new treatment strategies. Skp2 is a substrate-recruiting component of the E3 ubiquitin ligase complex, whose activity is regulated through neddylation. Slug is a transcriptional repressor involved in the epithelial-to-mesenchymal transition, which may contribute to therapy resistance. Although Skp2 has previously been associated with a mesenchymal phenotype and prostate cancer progression, the relationship with Slug deserves further elucidation. We have previously shown that a high Gleason score (≥8) is associated with higher Skp2 and lower E-cadherin expression. In this study, significantly increased expression of Skp2, AR, and Slug, along with E-cadherin downregulation, was observed in primary prostate cancer in patients who already had lymph node metastases. Skp2 was slightly correlated with Slug and AR in the whole cohort (Rs 0.32 and 0.37, respectively), which was enhanced for both proteins in patients with high Gleason scores (Rs 0.56 and 0.53, respectively) and, in the case of Slug, also in patients with metastasis to lymph nodes (Rs 0.56). Coexpression of Skp2 and Slug was confirmed in prostate cancer tissues by multiplex immunohistochemistry and confocal microscopy. The same relationship between these two proteins was observed in three sets of prostate epithelial cell lines (PC3, DU145, and E2) and their mesenchymal counterparts. Chemical inhibition of Skp2, but not RNA interference, modestly decreased Slug protein in PC3 and its docetaxel-resistant subline PC3 DR12. Importantly, chemical inhibition of Skp2 by MLN4924 upregulated p27 and decreased Slug expression in PC3, PC3 DR12, and LAPC4 cells. Novel treatment strategies targeting Skp2 and Slug by the neddylation blockade may be promising in advanced prostate cancer, as recently documented for other aggressive solid tumors.

## 1. Introduction

Prostate cancer (PCa) is the second most commonly occurring cancer as well as the second leading cause of cancer-related deaths in men in Western countries [1]. This malignancy develops as androgen-dependent and initially responds to androgen deprivation therapies; however, it ultimately progresses into a castration-resistant and largely incurable stage with metastases to the bones, lung, brain, or liver. Therefore, there is still an urgent need for better understanding of PCa progression to inspire discoveries and validation of new treatment strategies.

The process of prostate cancer metastasis and docetaxel treatment resistance is associated with epithelial-to-mesenchymal transition (EMT). EMT is an evolutionarily conserved transcriptional program, during which epithelial cells turn into a mesenchymal state. As a result, this increases their motility and invasive properties due to the loss of polarity and adhesion [2,3,4]. One of the hallmarks of EMT is the downregulation of E-cadherin, a marker of epithelial cells which may be conferred by Slug, a member of the Snail family of zinc-finger transcription factors [5]. Several reports described increased levels of Slug during carcinogenesis in different tumor types mainly associated with de-differentiation, relapse, metastasis, and poor prognosis [6,7,8]. We have also recently shown that Slug-expressing mouse prostate epithelial cells have increased stem cell potential [9].

Ubiquitination is one of the post-translational modifications of proteins and plays a crucial role in the ubiquitin–proteasome system (UPS) where the 26S proteasome degrades misfolded targeted proteins. This mechanism ensures homeostasis and proper functioning of cells [10,11]. The specificity and selectivity of the ubiquitin-conjugating system are conferred by more than 600 ubiquitin ligases (E3s), which are subclassified into three groups according to their mode of ubiquitin ligation [12]. The most abundant is the family of RING (really interesting new gene) E3 ligases. RING E3s rely on the enzymatic activity of E2s to ubiquitylate substrates and can act either independently or as a part of multi-subunit E3 complexes. An example of the latter is the family of cullin–RING E3 ligases (CRLs) whose activity is regulated through neddylation. In the process of neddylation, NEDD8 (neural precursor cell expressed, developmentally downregulated 8) molecule binds to cullin and releases the inhibitory interaction with CAND1 (cullin-associated and neddylation-dissociated 1). After dissociation of a polyubiquitinated substrate from CRLs, NEDD8 is detached by the COP9 (constitutive photomorphogenesis 9) signalosome for recycling [10,13].

S-phase kinase-associated protein 2 (Skp2) belongs to the family of substrate- recognition F-box proteins, which constitute one of the four subunits of CRL complex SCF (Skp1–cullin 1–F-box protein plus RING domain-containing component RBX1) [12]. Skp2 overexpression has been observed in various tumor types, including prostate cancer, and it is related to tumor progression and metastatic disease [11,14,15,16].

In our work, we confirmed an elevation of Skp2 in a cohort of patients with advanced prostate cancer and observed its correlation and coexpression with Slug. Coregulation of Skp2 and Slug was also observed in docetaxel-resistant prostate cancer cells and a mouse mesenchymal subline. A decrease of Slug was caused by MLN4924 (pevonedistat), an inhibitor of SCF^Skp2^ complex neddylation. Collectively, we report the coexpression of Skp2 and Slug in aggressive prostate cancer cells which can be inhibited by neddylation blockade, a potentially promising therapeutic approach for advanced prostate cancer.

## 2. Results

### 2.1. Expression of Skp2 Correlated with Slug in Patients with High Gleason Score and Metastasis to Lymph Nodes

We have previously shown that a high Gleason score is associated with higher Skp2 and lower E-cadherin expression in a retrospective cohort of 101 patients (Table 1) [17].

To investigate this interaction in more detail, we have extended this study for Slug, one of the negative regulators of E-cadherin, as well as for androgen receptor (AR), beta-catenin, and Ki67. Significantly increased expression of Skp2, AR, and Slug, along with lower expression of E-cadherin, was also observed in primary prostate cancer in patients who already had lymph node metastases (Figure 1). Skp2 was slightly correlated with Slug and AR in the whole cohort (Rs 0.32 and 0.37, respectively), which was enhanced for both proteins in patients with a high Gleason score (Rs 0.56 and 0.53, respectively) and, for Slug, also in patients with metastasis to lymph nodes (Rs 0.56; Table 2). Slug expression was also evaluated in tumor-adjacent areas of benign prostatic hyperplasia (BPH) from 32 patients. As reported by others [8,9,10,11,12,13,14,15,16,17,18,19], Slug expression was significantly higher in BPH in comparison to PCa (*p* < 0.001; median BPH histoscore 30, range 0–175).

### 2.2. Multiplex Immunohistochemistry Revealed Colocalization of Skp2 and Slug in Prostate Cancer Tissue

A more profound correlation between Slug and Skp2 in aggressive prostate cancer prompted us to perform dual staining in several prostate cancer tissues. We have indeed observed double-positive cancer cells, in particular in regions with high Gleason scores. The proteins were present in the same nuclei, although their signals did not entirely overlap (Figure 2). The image analysis was performed on the Mantra system (Appendix A), and the results were in good concordance with the results of the standard immunohistochemistry.

### 2.3. Mesenchymal Phenotype Is Associated with Enhanced Expression of Both Skp2 and Slug in Human PC3 and Mouse E2 Prostate Cancer Cell Lines

Next, we aimed to perform functional experiments, and we first made a screen of Skp2, Slug, and other proteins in three sets of prostate epithelial cell lines (PC3, DU145, and E2) and their mesenchymal counterparts. We have already reported that high Skp2 expression is associated with a mesenchymal phenotype and increased tumorigenic potential of DU145 prostate cancer cells [17]. These cells had a very low expression of Slug (Figure 3), which is in concordance with other studies [8]. On the other hand, docetaxel-resistant PC3 cells and mouse E2 cells presented mutual upregulation of Slug, Skp2, and mesenchymal markers in comparison to their epithelial counterparts (Figure 3).

### 2.4. Chemical Inhibition of Skp2, but Not RNA Interference, Modestly Decreased Slug Protein Levels

To assess the potential interaction between Skp2 and Slug, we employed both chemical and siRNA inhibition of Skp2. The high concentration of compound 25, a specific Skp2 inhibitor [20], decreased Slug expression to a lesser extent than Twist (Figure 4A). Despite a clear drop of Skp2 expression upon treatment with specific siRNA, we did not observe the downregulation of Slug (Figure 4B). Besides ubiquitination, Slug protein can also be regulated by acetylation [21]. However, we did not observe any change in Slug expression upon treatment with SIRT2 deacetylase inhibitor (Appendix A).

### 2.5. Neddylation Blockade Downregulated Slug Expression in PC3 and LAPC4 Cells

As mentioned above, Skp2 F-box protein is the substrate-recruiting component of the SCF (Skp1–cullin 1–F-box) type of E3 ubiquitin ligase complex whose activity is regulated by neddylation. When inhibited by MLN4924 (pevonedistat), we observed an apparent stabilization of p27 and decreased Slug expression in PC3 cells (Figure 5A). A recent study suggested FBXO11 as another E3 ligase responsible for Slug degradation upon MLN4924 treatment in uveal melanoma cells [22]. However, we did not observe any change in its expression upon the neddylation blockade (Figure 5A). Pevonedistat can also enhance the ubiquitination activity of MDM2, which has been reported to target both Slug and MDMX [23,24]. Nevertheless, we did not observe any change in MDMX levels in our cells upon MLN4924 treatment (Figure 5A). With respect to the crucial role of the androgen receptor in prostate cancer, we have performed additional experiments with several AR-positive cell lines (LNCaP, C4-2, LAPC4, and 22rv1; Figure 5B and Appendix A). As in the PC3 cells, we observed Slug downregulation upon neddylation blockade in LAPC4 cells, but not in C4-2 cells (Figure 5B). We have not observed any Slug expression in our authenticated 22rv1 cells. Importantly, we observed the downregulation of AR or its variants in all cell lines. The precise elucidation of Slug downregulation upon neddylation blockade in some prostate cancer cells will need further investigation.

## 3. Discussion

Enhanced expression of Skp2 had been repeatedly found in aggressive prostate cancer, which was also reproduced in our study [25,26]. We have also observed Skp2 correlation with the androgen receptor, in particular in patients with high Gleason scores. In a previous study, we showed that androgen depletion decreased prostate cancer cell proliferation, in part through downregulation of Skp2 [27]. Androgen regulation of Skp2 was also reported by others in LNCaP and C4-2 cells [28]; however, Skp2 can also promote prostate cancer independently of the androgen receptor [17,29,30].

We have also found a correlation between Skp2 and Slug, one of the EMT regulators. This correlation was stronger in tumors with higher Gleason scores in comparison to less aggressive tumors, as well as in patients with lymph node metastases in comparison to cases with localized disease. This is in line with the study by Esposito et al. who observed enhanced Slug expression along with pluripotency proteins Sox2 and Notch1 in high-grade PCa, lymph node metastases, and PCa with neuroendocrine features. The knock-down of Slug in PC3 downregulated the pluripotency genes as well as the neuroendocrine markers [8].

Of note, Slug expression is markedly modulated during carcinogenesis and cancer progression. Our analysis of the normal mouse prostate has recently shown that Slug+Trop+ cells possess a higher potential for organoid growth than Slug-negative populations [9]. Furthermore, we and others have found higher expression in BPH than in PCa either by immunohistochemistry [8] or mRNA expression profiling [18,19]. Esposito et al. demonstrated Slug-promoter methylation in neoplastic prostate epithelium, which was reversed at the invasion edge of high-grade PCa. Notably, Slug expression in aggressive cancer of various origins has repeatedly been reported [31,32,33].

As mentioned above, both Skp2 and Slug are associated with the process of EMT, which can contribute to chemotherapy resistance in various cancers [14,34]. Yang et al. reported both proteins upregulated in paclitaxel-resistant breast cancer cells and enhanced Skp2 in taxane-resistant prostate cancer. Skp2 also contributes to the castration-resistant phenotype via the stabilization of Twist, another EMT regulator [29]. We have reproduced Twist downregulation upon Skp2 inhibition with compound 25 in PC3 cells. However, Slug expression was only slightly decreased at the highest concentration of compound 25 and not significantly changed upon siRNA against Skp2.

The activity of the SCF^Skp2^ ubiquitin ligase complex is regulated by neddylation. Its inhibition by MLN4924 (pevonedistat) led to an apparent stabilization of p27 and decreased Slug expression in PC3, PC3 DR12, and androgen receptor positive LAPC4 cells. Pevonedistat has recently been suggested as a promising drug for metastasis inhibition via Slug downregulation in uveal melanoma [22]. The authors suggested E3 ligase FBXO11 as a regulator of Slug degradation upon MLN4924 treatment; however, we did not observe any change in its expression in our cell models. Pevonedistat was also reported to enhance the ubiquitination activity of MDM2, which targets both Slug and MDMX [23,24]. Nevertheless, we did not observe any change in MDMX levels upon the neddylation blockade in our cell models, except in C4-2 cells. Importantly, MLN4924 downregulated androgen receptor or its variants in all cell lines, which has also recently been reported by Zhou et al. [35]. Of note, they proved the synergistic effect of neddylation blockade with both enzalutamide and castration in vivo. Nevertheless, they also failed in the full elucidation of the neddylation-dependent inhibition of AR signaling. Slug expression can also be regulated by acetylation [21]; however, we again did not observe any change in Slug expression upon treatment with SIRT2 deacetylase inhibitor in PC3 and its docetaxel-resistant subline.

Important data will be obtained by future immunoprecipitation analysis of Skp2 and Slug. Another caveat of our study is the lack of cell fractionation analysis in in vitro experiments (i.e., Skp2 inhibition by siRNA, compound 25, and MLN4924) and the provision of only indirect evidence of Skp2 and Slug colocalization in cancer tissues and their coexpression in in vitro models. Follow-up data for patients analyzed by immunohistochemistry could also reveal an independent prognostic value of Skp2 and Slug. This study was focused on these two proteins; however, many other proteins (e.g., Twist 1, Zeb 1, vimentin) contribute to EMT, therapy resistance, and tumor progression. Monitoring their expression throughout our experiments would also improve the interpretation of the results and substantiate conclusions. Last but not least, Slug downregulation by neddylation blockade was not observed in all cell lines, and a deeper understanding of its cell-type-dependent regulation will be crucial for future therapies.

Although we did not find a direct mechanistic link between Skp2 and Slug, neddylation blockade downregulated Slug and inhibited Skp2, as documented by p27 upregulation, in several prostate cancer cells. Moreover, for the first time, we proved the occurrence of Skp2 and Slug in the same cancer cell nuclei in tumor tissue with high Gleason score and lymph node metastasis. Novel treatment strategies targeting Skp2 and Slug by neddylation blockade may be promising in advanced prostate cancer, as recently documented for other aggressive solid tumors [22,36,37].

## 4. Materials and Methods

### 4.1. Patients and Standard Immunohistochemistry

The study was approved by the Ethical Committee of the University Hospital and Faculty of Medicine and Dentistry, Palacky University in Olomouc (Ref. No. 127/14; approved on 21 August 2014). Archival formalin-fixed, paraffin-embedded prostate tumor samples obtained after radical prostatectomies between 1998 and 2011 (Table 1) were immunostained with appropriate antibodies according to standard manual and automatic techniques (Appendix A). Protein expression was assessed semiquantitatively by a pathologist using the histoscore method where the percentage of positive cells (0–100%) was multiplied by staining intensity (0–3), which resulted in a final histoscore between 0 and 300. Nuclear expression of Skp2 was considered in the main analyses; however, cytoplasmic expression was also observed to a lesser extent (Appendix AA). Predominant nuclear localization of Slug and expression of Skp2 both in nucleus and cytoplasm was confirmed by cell fractionation of the docetaxel-resistant PC3 cells (Appendix AB).

### 4.2. Multiplex Immunohistochemistry

Multiplex immunofluorescence staining was performed by Opal Fluorescent IHC Kit with tyramine signal amplification (Perkin Elmer, Beaconsfield, UK) on formalin-fixed paraffin-embedded tissue slices of selected cases according to the manufacturer’s protocol. Expression of Skp2 and Slug was monitored by Cy3 (Opal 570) and Cy5 (Opal 670), respectively. Cell nuclei were stained with DAPI. Images were captured by confocal microscope LSM780 and ZEN 11 software (Zeiss, Jena, Germany).

In addition, the slides were imaged using the MantraSnap 1.0.4 software included in Mantra Quantitative Pathology Workstation (Akoya Biosciences, Menlo Park, CA, USA) in DAPI, Cy3, and Cy5 acquiring filters. From each slide, five different regions of interest (ROI) were randomly selected, with a focus both on tumor parenchyma and stroma with 20 × 10 magnification. Images were analyzed using InForm 2.4.6. software (Akoya Biosciences, Menlo Park, CA, USA) with a prepared algorithm. This algorithm consists of linear unmixing and trainable steps of tissue segmentation, cell segmentation, and cell phenotyping and scoring. For tissue segmentation training, tumor, stromal, and empty areas were annotated by an experienced pathologist (MG). Each step was optimized on a set of different tumor pictures from the cohort and then applied to the batches. After automatic cell segmentation, the Skp2- and Slug-positive cells, as well as double-positive cells, were manually annotated and then applied to the batches. As a final step, cell scoring was performed. We set the positivity threshold both for Cy3 and Cy5 to be in agreement with the phenotyping step. We evaluated the percentage of single-positive, double-positive, and double-negative cells.

### 4.3. Cell Culture

PC3 and PC3 DR12 were obtained from Professor Watson (Dublin, Ireland) and cultured in DMEM (Dulbecco’s modified eagle medium; Sigma-Aldrich, St. Louis, MO, USA) supplemented with 10% FBS (fetal bovine serum) and antibiotics. PC3 DR12 cells were treated with a 12.5 nM concentration of docetaxel for two days every month [38]. E2 and cE2 murine prostate adenocarcinoma cells were kindly provided by Dr. Pradip Roy Burman (University of Southern California, Los Angeles, CA, USA) and cultured as previously described [39]. LAPC4 cells were obtained from Dr. Robert Reiter (University of Southern California, Los Angeles, CA, USA) and cultured in IMDM (Iscove’s Modified Dulbecco’s Medium; Sigma-Aldrich, St. Louis, MO, USA) supplemented with 10% FBS and 1nM R1881. LNCaP and DU 145 cells were obtained from ATCC (American Type Cultute Collection, Manassas, VA, USA) and cultivated in RPMI 1640 (Thermo Fisher Scientific, TFS, Waltham, MA, USA) with 10% FBS and antibiotics. The same culture conditions were used also for C4-2 and 22rv1 cells, which were kindly provided by Dr. Marian Hajduch (Palacky University Olomouc, Czechia) and Dr. Culig (Innsbruck Medical University, Austria), respectively. The epithelial and mesenchymal sublines were established from sorted single-cell clones derived from Trop-2-positive or Trop-2-negative DU145 populations, as previously described [40]. The AmpFLSTR Identifiler PCR Amplification Kit (ThermoFisher, Waltham, MA, USA) was used to verify the origin of cell lines [17].

### 4.4. Western Blot Analysis and Immunodetection

Protein extraction and immunoblotting were performed as previously described [41]. Briefly, cells were harvested in a RIPA (Radio-Immunoprecipitation Assay) buffer supplemented with Protease/Phosphatase Inhibitor Cocktail (Roche, Basel, Switzerland). Twenty micrograms of whole-cell lysate was separated by electrophoresis in 10% Bis-Tris polyacrylamide gel and transferred by semidry blotter (BioRad, Hercules, CA, USA) on a nitrocellulose membrane (GE Healthcare, GE Healthcare, Chicago, IL, USA). Selected proteins were recognized by specific primary antibodies (Appendix A) and horseradish peroxidase linked secondary antibodies. Chemiluminescent signals were captured on Fc Odyssey Imaging system (LI-COR, Lincoln, NE, USA) upon the addition of SuperSignal West Dura or Femto substrates (ThermoFisher, Waltham, MA, USA).

### 4.5. Chemical Inhibitors

Inhibitors of Skp2 (compound 25, alias SZL-P1-41; range 1.25–20 μM; Tocris, Bristol, UK), histone deacetylases (sirtinol; range 1.56–20 μM; Tocris), and neddylation (MLN4924, pevonedistat; range 0.04–2 μM; Active Biochem, Hong Kong) were first tested by MTT cell viability assay, as described previously [42]. PC3 and PC3 DR12 cells were treated with a concentration scale (10–40 μM) of compound 25 and incubated for 48 h. After 24 h of incubation of cells with 25 μM sirtinol, cycloheximide (50 μg/mL) was added for 2 and 4 h. PC3, PC3 DR12, and C4-2 and 22rv1 cells were also treated with 2 μM of MLN4924 (pevonedistat) and incubated for 24 h. LAPC4 cells were treated in the same way but with 1 μM of MLN4924. All experiments were performed at least three times.

### 4.6. RNA Interference

PC3 and PC3 DR12 cells were treated with Lipofectamine RNAiMAX (Invitrogen, Carlsbad, CA, USA) with a 20 nM concentration of control siRNA (UAA UGU AUU GGA ACG CAU A; Eurofins Genomics, Luxembourg) or a 20 nM mixture of endoribonuclease-prepared anti-Skp2 siRNAs (EHU094381, Sigma Aldrich) for 48 h. Lower concentrations (down to 0.5 nM) resulted in less efficient Skp2 downregulation. Experiments were performed at least three times.

### 4.7. Cell Fractionation

PC3 DR12 cells were lysed 1 h on ice in the hypotonic buffer (50 mM Hepes, pH 7.3, 10 mM potassium chloride, 2 mM magnesium chloride, 1 mM dithiothreitol, 0.1% NP40 and Protease Inhibitor Cocktail (Roche, Basel, Switzerland)) by subsequent homogenization with 25G syringes. Cytoplasmatic fraction was obtained as supernatant after 15 min of centrifugation at 1000× *g* and 4 °C. Nuclear fraction was obtained from the pellet by addition of RIPA buffer supplemented with Protease/Phosphatase Inhibitor Cocktail (Roche, Basel, Switzerland) [43]. Experiments were performed three times.

### 4.8. Statistical Analysis

The protein expression data were analyzed with respect to the clinical–pathological parameters (serum PSA, Gleason score, tumor stage, and lymph node status) by Kruskal–Wallis test, Mann–Whitney test, Wilcoxon signed-rank test, and Spearman’s rank correlation coefficient using the program Statistica 12 (TIBCO Software Inc., Palo Alto, CA, USA). Graphs were generated in GraphPad Prism 8 (GraphPad Software, San Diego, CA, USA).

## Figures and Tables

**Figure 1 ijms-22-02844-f001:**
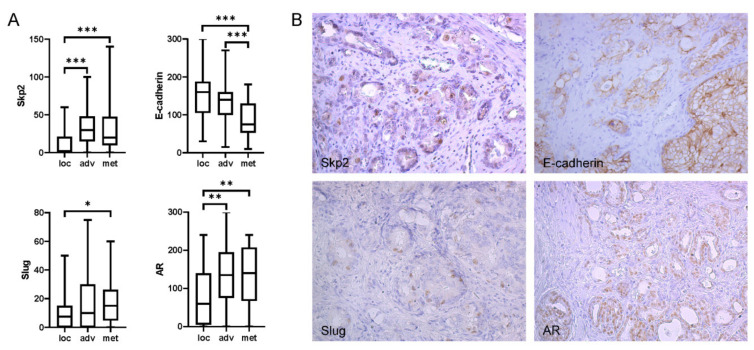
Immunohistochemical analysis of Skp2, AR, Slug, and E-cadherin. (**A**) Significantly increased expression of Skp2, AR, and Slug, along with lower E-cadherin expression, was observed in primary prostate cancer in patients who already had lymph node metastases (met). Localized (loc) and advanced (adv) tumors were defined as “pT2a-c, N0” and “T3–T4, N0”, respectively. Box-plots represent median, 25–75% percentiles, and range of values. *p*-values < 0.05, <0.01, and <0.001 are indicated by *, **, and ***, respectively. (**B**) Example of immunohistochemical staining of Skp2, E-cadherin, Slug, and androgen receptor in primary prostate cancer tissue with Gleason score 8 (4 + 4). Another example of the staining for localized prostate cancer (PCa) is provided in Appendix A (all images are at magnification ×200).

**Figure 2 ijms-22-02844-f002:**
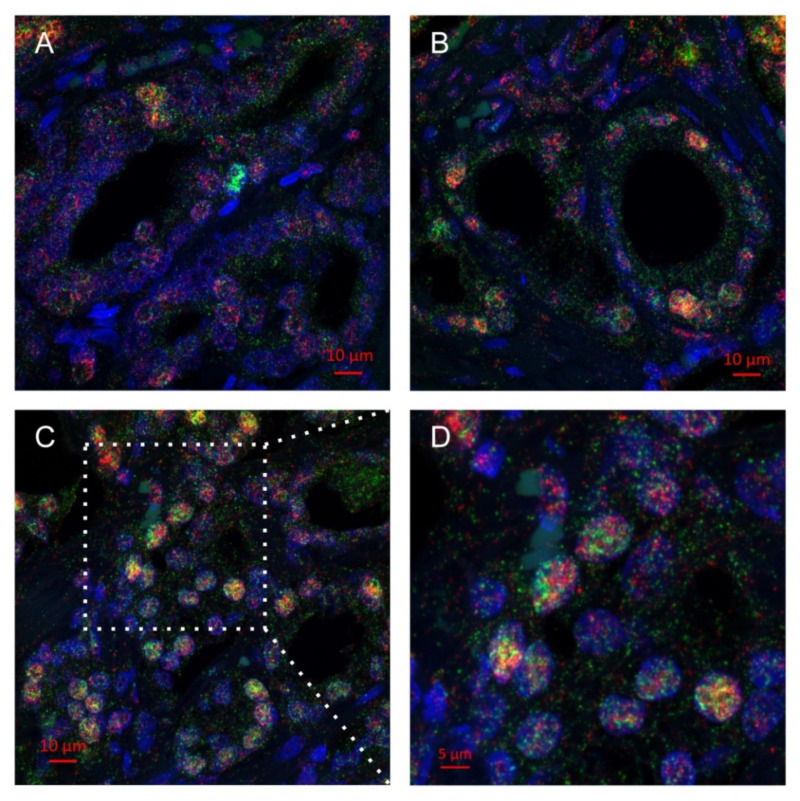
Multiplex immunohistochemistry of Skp2 (green) and Slug (red) in primary tumors from two patients with metastasis to lymph nodes. (**A**) Occasional double-positive tumor nuclei (6%, 4/40) were observed in the tissue with Gleason grade 3 from patient #21 (pT2c, pN1, Gleason score 3 + 2). (**B**) Similar frequency of colocalization (10%, 4/40) was also observed in the Gleason grade 3 tissue from another patient, #62 (pT3a, pN1, Gleason score 4 + 3), while more frequent double-positive nuclei (19%, 11/57) were found in the area of Gleason grade 4 tissue (**C**,**D**) from this patient. Nuclei were stained with DAPI (blue; 4′,6-diamidino-2-phenylindole dihydrochloride). Images were captured by confocal microscope LSM780 (Zeiss).

**Figure 3 ijms-22-02844-f003:**
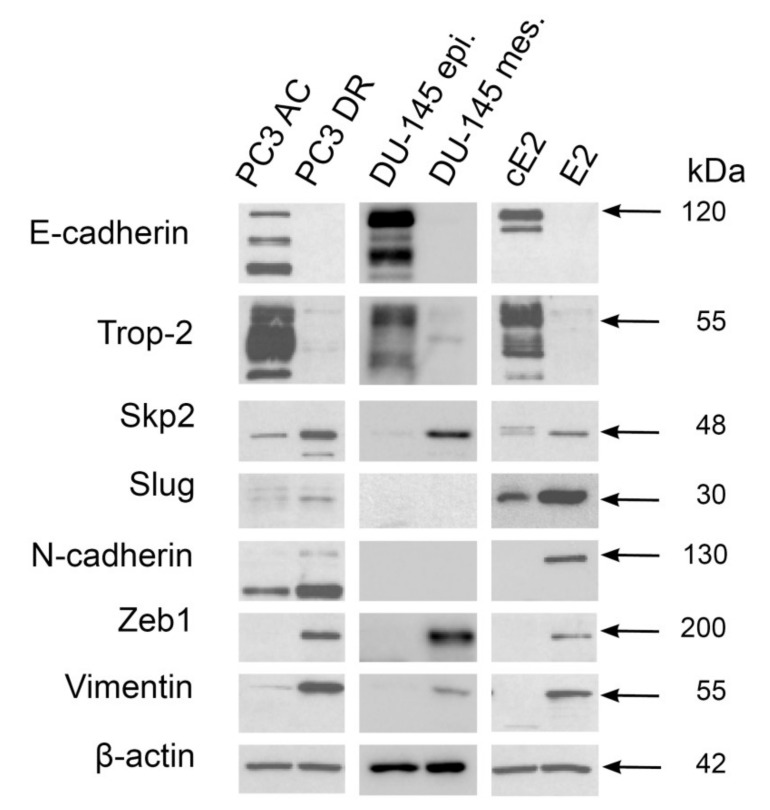
Western blot analysis of Skp2, Slug, and other selected proteins associated with epithelial or mesenchymal phenotype in prostate cancer cell lines. Epithelial cells (PC3 AR, DU145, and cE2) and their mesenchymal counterparts (PC3 DR, DU145 mes., and E2) were analyzed in three repetitions. All tested cell lines showed upregulation of Skp2 in cases of mesenchymal phenotype. Slug upregulation was detected in PC3 DR and E2, but not in DU145 mes. cells. Other epithelial (E-cadherin and Trop-2) and mesenchymal (N-cadherin, Zeb1, and vimentin) markers showed upregulation within the cells with the corresponding phenotype, except for N-cadherin in DU145 cells, presumably because of very limited levels of this protein.

**Figure 4 ijms-22-02844-f004:**
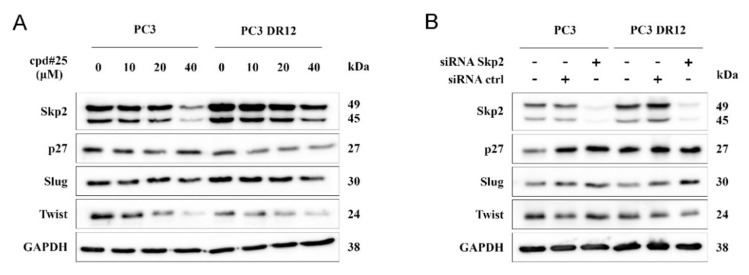
Chemical inhibition of Skp2, but not RNA interference, modestly decreased Slug protein levels. (**A**) The high concentration of compound 25, a specific Skp2 inhibitor, decreased Slug expression but to a lesser extent than Twist. (**B**) Despite a clear drop of Skp2 expression upon treatment with specific siRNA, we did not observe downregulation of Slug PC3 and PC3 DR12 cells.

**Figure 5 ijms-22-02844-f005:**
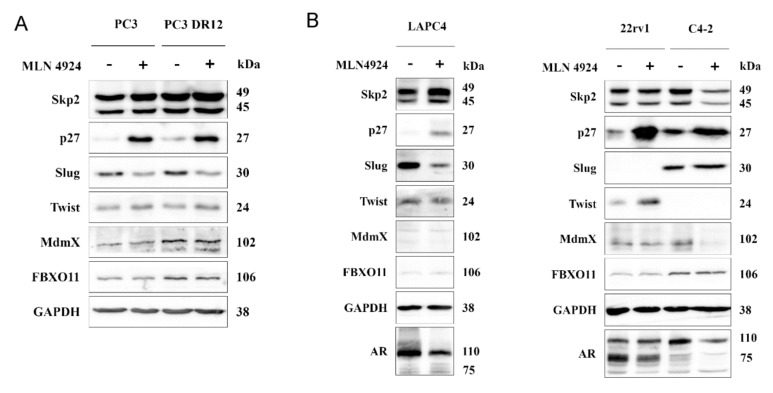
Neddylation blockade downregulated Slug expression. (**A**) When inhibited by MLN4924 (pevonedistat), we observed an apparent stabilization of p27 and decreased Slug expression in PC3 cells as well as in the docetaxel-resistant PC3 DR12 cells. (**B**) Slug was also downregulated upon neddylation blockade in androgen receptor positive LAPC4 cells, but not in C4-2 cells. We have not observed any Slug expression in our authenticated 22rv1 cells.

**Table 1 ijms-22-02844-t001:** Clinical characteristics of retrospective prostate cancer patient samples (N = 101).

Characteristics	Subgroups	N	%
Age	49–60	36	35.6
	61–70	57	56.4
	71–76	8	7.9
Serum PSA (ng/mL)	<4	12	11.9
	4–10	40	39.6
	>10	48	47.5
	missing	1	1.0
Gleason score	<7	22	21.8
	7	49	48.5
	>7	30	29.7
Cancer stage	pT2a-c	42	41.6
	pT3a-b	50	49.5
	pT4	9	8.9
Lymph node status	pN1	29	28.7
	pN0	65	64.4
	pNx*	7	6.9

* Without lymphadenectomy.

**Table 2 ijms-22-02844-t002:** Spearman correlations (Rs) of selected proteins in the whole cohort and patients with a high Gleason score or positive lymph nodes.

Protein 1	Protein 2	Whole Cohort (N = 101)	Gleason > 7 (N = 30)	Positive LN (N = 29)
Skp2	Slug	0.322	0.557	0.559
	androgen receptor	0.370	0.535	0.372
Slug	androgen receptor	*0.154*	0.374	*0.285*
Ki-67	androgen receptor	*0.054*	*0.246*	0.504
	beta-catenin membrane	*−0.193*	−0.234	−0.418
E-cadherin	beta-catenin membrane	*0.106*	0.423	0.615

Rs coefficients above 0.5 are highlighted in bold, while insignificant values are in italics.

## Data Availability

All important data generated or analyzed during this study are included in this article (and its Appendix A).

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
