# Peer review of "Skp2 and Slug Are Coexpressed in Aggressive Prostate Cancer and Inhibited by Neddylation Blockade"

_ijms, 2021, doi:10.3390/ijms22062844_

Round 1

Reviewer 1 Report

This manuscript aimed at illustrating the relationship between Skp2 and Slug in prostate cancer. Even though the authors have added some narratives and supplements some experimental results in this revised manuscript. However, the authors did not make a point-by-point response based on this reviewer's suggestions, including:

  1. According to their immunohistochemical analysis of Skp2, Androgen receptor, Slug and E-cadherin in Fig.1, the result showing the expression of Skp2 was correlated with Slug, Androgen receptor, and E-cadherin in related human primary prostate cancer tissues. However, there is still no reasonable way to explain why they only focus on these EMT-related factors on prostate motility and aggressiveness activity.

  2. Still, no comparable data to claim the importance of Slug from the relationship of Skp2 and Vimentin as considering their previous studies (doi: 10.1038/s41598-019-42131-y), which one is more important for Skp2 mediated prostate cancer malignancy? Besides, according to the supplementary figure.4, the data shows Skp2 expression was independent of Androgen receptor and did not correlate to EMT gene expressions. Of note, the androgen receptor was correlated to Slug expression, why? Moreover, using only one androgen receptor-dependent cell is not enough to explain whether the androgen receptor is not related to Skp2 in supplementary figure.4; what is the related expression of DU145? More importantly, the authors cannot rule out the importance of the androgen receptor, according to many studies related to Skp2 and androgen receptor.

  3. The authors must provide appropriate reasons to convince how the inconsistency result as compared to their immunohistochemical analysis. Significantly, according to their multiplex, IHC results in Fig.2, the expression of Skp2 is primary in the nucleus. There is no relative analysis to provide the corresponding subcellular distribution of each factor correlated to Skp2 expression in the cytosol and nucleus in the supplementary figure.4. Therefore, the immunoprecipitation-related analysis must be performed to demonstrate Skp2 and Slug's direct interactions in the nuclei to support the multiplex IHC results. The authors must learn an important concept that co-localization does not mean there is an interaction that has happened.

  4. Scientific research must be verified using the two-way model; according to the current statement, there is no comparative way to point out that the main downstream effector of Skp2 is P27 and whether P27 can also be seen in immunohistochemical analysis consistent with Skp27 still unknown. Besides, there may be concerns about the side effects caused by barely using a siRNA or inhibitors for experiments. More importantly, a dose-dependent way is necessary as well as the overexpression and knockdown experiments should be included in this manuscript.

  5. More is needed because this article does not provide any related biological assays to link to the clinical outcome. According to the degradation involvement of Skp2 on Slug, supplementary figure.2 did not provide any direct evidence showing any relationship with Skp2. Whether it is related to Neddylation, there are many doubts at present, even in the inhibition of Neddylation downregulated Slug expression, what is the direct involvement of Skp2? Furthermore, the past research found that Skp2 degradation is affected by neddylation inhibitor; why is it inconsistent with the result in Fig.5?

  6. The authors has not provided any further experimental-based results of immunohistochemistry staining and immunofluorescence analysis to explain. According to the current statement, there is no comparative way to point out that the main downstream effector of Skp2 is P27 and whether P27 can also be seen in immunohistochemical analysis consistent with Skp27 still unknown.

  7. Although the authors have supplemented the data, there are problems including: There are multiple bands as non-specific signals were represented in the immunoblotting assay, which one is the major one? The authors must point out and must be quantified. In Fig.4B, the inconsistent results were shown by knockdown control that can decrease Twist and can be increased in the Skp2 knockdown group. The similar in-consistent results can be observed in Fig.5; the Slug was decreased in MLN treatment but increased in the Skp2 knockdown group (Fig.4). The Slug and Twist expression was representing an in-consistency response to MLN4924 treatment of different cells. The poor Nuclear and cytoplasmic separation can be observed in supplementary Fig. 3, resulting in the inability to judge its authenticity.

  8. The reference or by using "beyond the scope of their research" did not provide a good reason to explain why the authors are reluctant to perform the related necessary experiments per requested by this reviewer. These questions are essentially and critically raised according to this study which are feasible and reasonable.
  9. Finally, in order to conduct scientific research logically, the authors should repeat the questions previously raised by the reviewer to increase the rationality of many conclusions in this research. The revised manuscript is still not enough to answer many critical questions raised.

Reviewer 2 Report

The manuscript from Mickova and colleagues describes an interesting study carried out on prostate cancer (PC) human samples and several PC cell lines, showing the concomitant expression of the S-phase kinase-associated protein 2 (Skp2) and Slug in aggressive PC, and a potential therapeutic approach aiming at decreasing these pro-metastatic factors by inhibiting neddylation. This strategy has been successfully tested in different tumor types (e.g. melanoma); here, the authors provide preliminary evidence of the promising effectiveness of neddylation inhibitors (i.e. MLN4924) in the treatment of metastatic PC. Far from being an exhaustive study on the interplay between Skp2 and Slug in PC progression and metastasis, the authors report a list of experiments that suggest an independent behavior of the two proteins in the context of PC. Though, they found a common mechanism regulating the expression of both Skp2 and Slug - neddylation - that could be targeted in order to decrease/revert the epithelial-mesenchymal transition (EMT) driven by these proteins. This work represent an interest read for people working on EMT in PC and bargains several seed for future researches.

Excluding the abstract, that need to be strongly improved, the manuscript is clearly and exhaustively written, methodologies are well described, the experimental system is adequate and verified. The literature background, the language used and the structure of the paper match the quality requests of the journal.

In order to be accepted for publication, the authors should address some minor issues that undermine the strength of the work:

  • The abstract should be drastically improved. In the present form, the authors listed a mere sequence of the experiments they carried out with too many unnecessary technical details (like all the cell lines used for each experiments, or the values of Gleason scores). Neddylation inhibition pops up in the text without describing its rationale and the link with Skp2 and Slug. It is not easy to identify the aim of the research and the main findings from the abstract. I suggest a complete re-writing.
  • In Figure 1A legend it is not specified what “loc” “adv” and “met” means. In figure 1B E-cadherin is reported as CDH1 in the figure, although not specified in the legend.
  • In figure 1B the authors should provide a representative picture also of non-tumor or not metastatic PC immunohistochemical staining – a representative image of what is called and quantified “loc” in figure 1A.
  • In figure 2, the authors should quantify in each field acquired (or at least in the ones showed) the percentage (%) of nuclei in which Skp2 and Slug colocalize.
  • In figure 3, the immunoreactive band of N-cadherin in E2 cells seems to be flawed. The authors should replace this WB with a better representative.
  • Ln56: “One of the hallmarks of EMT is inhibition of E-cadherin..” it would be more correct “downregulation” or “E-cadherin expression decrease” than inhibition.
  • Check in the whole text if acronyms are extended at first appearance in the text (e.g. NEDD8 in ln71, COP9 in ln74) and uniform the nomenclature (Nedd8 or NEDD8).

Reviewer 3 Report

The present manuscript is a manuscript about the association of Skp2 and Slug in aggressive prostate cancers and their inhibition by neddylation blockade. The topic is timely, even if not very novel. The authors analyzed the expression of the proteins (Skp2, Slug, AR, Ki-67, beta-catenin, and E-cadherin) in the prostate cancer patient cohort by IHC and multiplex IHC staining. They found a correlation between Skp2 and Slug, especially their correlation enhanced in patients with high Gleason score and metastasis to lymph nodes. Western blot analysis confirmed these proteins’ expression in prostate cancer cells with AR-positive and negative, and the correlation of Skp2 and Slug. By comparing the results between using RNA interference and the Skp2 inhibitor, and the neddylation blocker, the authors confirmed that neddylation blocker inhibited Skp2 and upregulated p27 and thus, decreased Slug expression. Therefore, they concluded that the two proteins (Skp2 and Slug) had a possible link, and targeting them by neddylation blockade may be a novel treatment strategy. The manuscript is overall clearly written, easy to understand.

The below comments should be addressed to strengthen this manuscript before its publication.

Major comments:

  1. This article would benefit from close editing.
  2. In figure 5, the authors provided proof that neddylation blockade stabilized p27 and decreased Slug in PC3 and PC3 DR12 cells. But lack of an experiment on how p27 directly inhibits Slug protein. Please add an experiment to show the direct relationship between p27 and Slug.
  3. The references in the text are numbered, but in the References section are not numbered. This increases the difficulty to check the references.
  4. The figures’ insertion in the text is not reasonable. There is a figure under the introduction section!

Minor comments:

  1. In line 22, Prostate cancer “patient’s cohort”, should be prostate cancer patient cohort.
  2. In lines 31-32, “….prostate cancer of patients who had….” should be “…prostate cancer in patients who had….”.
  3. In line 27 and other lines: “in-vitro” at more situations, used like in vitro.
  4. In table 1: Column %, all the numbers use a comma instead of a point. Please correct.

Round 2

Reviewer 1 Report

This article aims to link the relationship of Skp2 and Slug in the aggressive prostate cancer. In this revised manuscript, this reviewer asked the following questions based on the author’s point-by-point response and careful evaluation:

  1. In this article, the authors tried to link the higher Skp2 and lower E-cadherin expression in prostate cancer with a high Gleason score (̤≥8). Based on past findings and proposed, Slug is one of the negative E-cadherin regulators to rationalize Skp2 and Slug's relationships. However, according to the author's results, Skp2 was slightly correlated with Slug and AR in the whole cohort (Rs 0.32 and 0.37, respectively). Although the androgen receptor is also one of Slug's negative regulators, it only showed a "slight" trend. Such a phenomenon cannot obtain significant results in in vitro assays (Supplementary Figure 4), even though Skp2 and Slug's relationship had not been investigated before.
  2. According to the authors published articles (Sci Rep. 2019 Apr 5;9(1):5695. doi: 10.1038/s41598-019-42131-y.) as previously described: 1. a high Gleason score was associated with a mesenchymal phenotype, as demonstrated by decreased E-cadherin expression and increased vimentin expression, 2. the induction of mesenchymal phenotype was defined by the expression of Vimentin, N-cadherin and transcription repressor of E-cadherin, ZEB1.), 3. in our work, we showed that Skp2 expression was increased in cancerous prostate tissue compared to benign prostate tissue, specifically in tumors with a high Gleason score (≥7). Poorly differentiated tumors also had lower expression levels of E-cadherin and higher levels of Vimentin, suggesting the ongoing EMT of prostate cancer cells.) Although the authors found a correlation of Vimentin neither with Skp2 nor Slug. However, in this article, the authors still used Vimentin to observe the epithelial or mesenchymal phenotype. (Fig.3) Although the results in Fig.3 also showed that high SKP2 is related to Zeb1 and Vimentin, the authors only used Slug as their follow-up research. By the way, Zeb1 and Vimentin in PC3 DR, DU145 mes, and E2 have a co-expression trend with Skp2, but the authors only compared the differenced between Twist and Slug in other experiments. How about the expression of twist in these cells (Figure.3) and the related expression of Zeb1 and Vimentin in supplementary Figure 4? Otherwise, there are no reasons to use Twist and Slug only to compare the difference after Skp2 changes (Fig.4,5,). The current results could be troublesome and can not stand through the checking by others. The authors need to integrate these differences throughout the manuscript and correct them as this reviewer's question 5 has asked.
  3. Regarding with the issues related to the androgen receptor, first of all, the reviewerd did not deny whether Skp2 or Slug is a potential future therapeutic target for neddylation inhibition. However, there are still many essential issues that need to be presented to link the Skp2 and Slug relationship and the androgen receptor in this article. The related literatures and the neddylation blockade results (Fig.5b) provided by authors still can not convince this reviewer, especially, the related side effects and whether they can be reversed through the over-expression of skp2 under compound 25 (Fig.4) or MLN4924 (Fig.5) treatment remain doubtful.
  4. The authors must provide appropriate reasons to convince this reviewer how the inconsistency results occurred as compared to their immunohistochemical analysis one. Significantly, according to the author's multiplexed IHC results in Fig.2, the expression of Skp2 is primary in the nucleus. There is no relative analysis to provide the corresponding subcellular distributions of each factor correlated to Skp2 expression in the cytosol and the nucleus in the supplementary figure.4. Therefore, the immunoprecipitation-related analysis must be performed to demonstrate Skp2 and Slug's direct interactions in the nuclei to support the multiplex IHC results accuracy. The authors must learn an important concept that co-localization does not mean there is an protein-protein interaction that has happened.
  5. Scientific research must be verified by using the two-way model. According to the current statement, there is no comparative way to point out that the main downstream effector of Skp2 is P27 and whether P27 can also be seen in immunohistochemical analysis consistent with Skp27 still unknown. Besides, there may be concerns about the side effects caused by using a siRNA or inhibitor for experiments. More importantly, a dose-dependent way is necessary as well as the overexpression and knockdown experiments should be included in this manuscript.
  6. More is needed because this article does not provide any related biological assays to link to the clinical outcomes. According to the degradation involvement of Skp2 on Slug, supplementary figure.2, the authors still did not provide any direct evidence showing any relationship with Skp2. Whether it is related to Neddylation, there are still many doubts left at present, even in the inhibition of Neddylation downregulated Slug expression, what is the direct involvement of Skp2? Furthermore, the past researches have found that Skp2 degradation is affected by neddylation inhibitor. Why is it inconsistent with the result in Fig.5?
  7. This reviewer can understand that many important questions cannot be answered without  experiment results performed and the solid results were obtained one after another in a short time. However, the authors still needs to provide full replies to these questions raised by this reviewer in Introduction, Results, and Discussion section in in meaningful paragraphs, including many logical question's answers. Importantly, the comparisons of the other published paper's results with this study to show the merits and new findings are necessary and should be written in paragraphs in Discussion . In the reviewer's questions that the authors can not provide any further experimental results due to COVID-19 pandemic or time limitation to back up the authors claims, this study's limitations, future plans, and potential impact/significance must be provided in Discussion. 

Reviewer 3 Report

The authors have reasonably answered questions incurred in the previous version. This version should be acceptable.

Author Response

Dear Reviewer, thank you for the positive evaluation. We are glad that you consider our last version of the manuscript suitable for publication. With kind regards, Jan Bouchal and on behalf of co-authors

This manuscript is a resubmission of an earlier submission. The following is a list of the peer review reports and author responses from that submission.

Round 1

Reviewer 1 Report

I like the ms:  Skp2 and Slug are coexpressed in aggressive prostate cancer and
inhibited by neddylation blockade by Mickova, et al. for several reason:

a-It is clearly and simply written in all its part. The authors privileged an essential style without lingering in disturbing and confusing elements, and this shows, in my view, that they have a sound knowledge of the topic.

b-The experimental design is consistent with the author hypothesis;

c-The experiments are well done and the controls are appropriate;

d-the results are clearly described;

e-the discussion  is articulate and conclusions are honest and fair.

Reviewer 2 Report

The study by Mickova et al. intend to identify the roles between Skp2 and Slug in prostate carcinogensis. The authors found that the expression of Skp2 and Slug correlated with Gleason score and metastasis phenotypes in the prostate cancer patients. They showed that the expression of Skp2 and Slug were co-localized in the nuclei and associated with the mesenchymal phenotype. Moreover, inhibition of Skp2 by specific inhibitor decreased Slug expression and increase tumor suppressor P27 expression through the E3 ubiquitin-ligase complex neddylation. However, this manuscript needs to add additional descriptions and analyzed results to convince the general International Journal of Molecular Sciences readerships of the author's hypothesis. Several issues must be addressed.

Major:

  1. In Fig.1, the authors provided evidences that expression of Skp2 correlated with Slug, E-cadherin, and androgen receptor in the primary prostate cancer tissues. However, there are multiple EMT-related factors involved in prostate motility and aggressiveness. The authors must give a reasonable explanation to link Skp2 and Slug. Besides, according to the author's previous published paper (doi: 10.1038/s41598-019-42131-y), they have already demonstrated that the expression of Skp2 was correlated with vimentin. One would wonder which factor is more important for the Skp2-mediated prostate cancer malignancy. The authors must provide appropriate reasons to convince that Slug is the most effective factor in Skp2-mediated tumor progression in prostate cancer metastasis.

  1. In Fig.2, according to the multiplex IHC results, high Skp2 is expressed in nucleus. The authors must fractionate the subcellar components to present that Skp2 and EMT markers expressions to distinguish the differences between the nucleus and cytosol expressions in a panel of prostate cancer cell lines throughout this manuscript.

  1. Prostate cancer can be distinguished from progression to androgen-dependent or androgen-independent. The authors must provide the evidences to distinguish the correlations of Skp2 in a panel of prostate cancer cell lines. Based on the current model and results, Skp2 expressions should be correlated with androgen-receptor expression in human clinical samples (Fig.1). Besides, emerging evidence has shown that androgen receptor involved in the regulation of Skp2. However, PC-3 and DU145 cell lines are androgen receptor-independent. The authors must use androgen receptor-dependent cell lines for the in vitro experiments throughout the manuscripts.

  1. In clinical part, the authors must base on the components of Skp2, AR, Slug, and E-cadherin as well as the P27, E3 ubiquitin-ligase complexs to perform the clinically relevant in prostate cancer association analyses, such as the first progression, overall survival, disease-free survival, and post-progression survival in the different conditions of patients' cases in a systematic way.

  2. In Fig.2, according to the multiplex IHC results. The author must provide additional experiments to support and demonstrate the interactions of Skp2 and Slug in the nucleus. It has been shown to be able to stain two nuclei expressed protein in the same subcellular components. The authors must follow this trait. The experimental approaches and the results presented in this current manuscript are not satisfying and rather weak. 

  1. The authors must perform the overexpression and knockdown experiments to verify and validate Skp2 or interested molecules inducing the cell motility or malignancy potentials. Current experiments are not vigorously enough to convince or support the correlation of Skp2 and Slug involved in prostate metastasis.

  1. Previous reports demonstrate that Skp2, as a receptor protein for 26S proteosome degrade target molecules. The authors must perform additional experiments, such as degradation assay or use inhibitors to demonstrate that Slug is involved in Skp2-mediated protein stability or degradation. Current experimental results are not enough to convince or support the authors' proposed model.

  1. In Figure 5, the authors provided evidences that the inhibition of neddylation downregulated Slug expression. The authors must validate Skp2 mediated E3 ubiquitin-ligase complex is involved and required for the neddylation. One of the concerns is that treating inhibitors of neddylation may provide side-effects and not specific for Skp2. The authors must perform key experiments to identify which molecule is involved in the neddylation-mediated Slug degradation in prostate cancer cells. Moreover, previous reports demonstrate that neddylation inhibitor can induce Skp2 degradation. The author must explain why the inconsistent results of Skp2 expression in Fig.5.

  1. The authors showed that E3 ubiquitin-ligase complex neddylation involved in Skp2-mediated Slug degradation (Fig.5). The immunohistochemistry staining and immunofluorescence staining analyses must be performed in these sections to validate the component involved in E3 ubiquitin-ligase complex-mediated degradations in prostate cancer tissues.

Minor:

  1. In Figure 3, the DU145 cell seems to be with no Skp2 expression. These experiments must be performed with another prostate cancer cell line.

  1. Skp2 has multiple downstream targets, such as P21, P27, P57, P130, BRCA2, ATF4. The authors must add other downstream targets to point out that P27 is the most effective functional target.

  1. siRNA and inhibitors treatment must provide dose-dependent effect throughout the manuscript.